# Sex differences in bile acid homeostasis and excretion underlie the disparity in liver cancer incidence between males and females

Megan E Patton[1], Sherwin Kelekar[1], Lauren J Taylor[1], Angela E Dean[1,2], Qianying Zuo[3], Rhishikesh N Thakare[4], Sung Hwan Lee[5,6], Emily C Gentry[7†], Morgan Panitchpakdi[7], Pieter Dorrestein[7], Yazen Alnouti[4], Zeynep Madak-Erdogan[2,3,8], Ju-Seog Lee[5], Milton J Finegold[9], Sayeepriyadarshini Anakk[1,2,8]*

[1]Department of Molecular and Integrative Physiology, University of Illinois, Urbana-Champaign, Urbana, United States; [2]Division of Nutritional Sciences, University of Illinois, Urbana-Champaign, Urbana, United States; [3]Department of Food Science and Human Nutrition, University of Illinois at Urbana-Champaign, Urbana, United States; [4]Department of Pharmaceutical Sciences, College of Pharmacy, University of Nebraska, Omaha, United States; [5]Department of Systems Biology, The University of Texas MD Anderson Cancer Center, Houston, United States; [6]CHA Bundang Medical Center, CHA University School of Medicine, Seongnam, Republic of Korea; [7]Collaborative Mass Spectrometry Innovation Center, Skaggs School of Pharmacy and Pharmaceutical Sciences, University of California San Diego, San Diego, United States; [8]Cancer center at Illinois, University of Illinois, Urbana-Champaign, Urbana, United States; [9]Department of Pathology, Baylor College of Medicine, Houston, United States

*For correspondence: anakk@illinois.edu

Present address: [†]Department of Chemistry, Virginia Tech, Blacksburg, United States

## eLife Assessment

This study provides **valuable** insights into the influence of sex on bile acid metabolism and the risk of hepatocellular carcinoma (HCC). The data to support that there are inter-relationships between sex, bile acids, and HCC in mice are **convincing**, although this is a largely descriptive study. Future studies are needed to understand the interaction of sex hormones, bile acids, and chronic liver diseases and cancer at a mechanistic level. Also, there is not enough evidence to determine the clinical significance of the findings given the differences in bile acid composition between mice and men.

**Abstract** Hepatocellular carcinoma (HCC), the common liver cancer, exhibits higher incidence in males. Here, we report that mice lacking bile acid (BA) regulators, Farnesoid X Receptor (FXR also termed NR1H4) and Small Heterodimer Partner (SHP also termed NR0B2), recapitulate the sex difference in liver cancer risk. Since few therapeutic options are available, we focused on understanding the intrinsic protection afforded to female livers. Transcriptomic analysis in control and NR1H4 and NR0B2 double knockout livers identified female-specific changes in metabolism, including amino acids, lipids, and steroids. To assess translational relevance, we examined if transcriptomic signatures obtained from this murine HCC model correlate with survival outcomes for

HCC patients. Gene signatures unique to the knockout females correspond with low-grade tumors and better survival. Ovariectomy blunts the metabolic changes and promotes liver tumorigenesis in females that, intriguingly, coincides with increased serum bile acid (BA) levels. Despite similar genetics, knockout male mice displayed higher serum BA concentrations, while female knockouts excreted more BAs. Decreasing enterohepatic BA recirculation using cholestyramine, an FDA-approved resin, dramatically reduced the liver cancer burden in male mice. Overall, we reveal that sex-specific BA metabolism leading to lower circulating BA concentration protects female livers from developing cancer. Thus, targeting BA excretion may be a promising therapeutic strategy against HCC.

## Introduction

Liver cancer, a leading cause of cancer-related death, has diverse etiologies and displays sex-difference with reduced risk in females compared to males (*El-Serag, 2012*; *White et al., 2017*; *GBD 2013 Mortality and Causes of Death Collaborators, 2015*; *Forner et al., 2012*; *Llovet et al., 2021*). Since current therapies for liver cancer fall short, we posit that understanding molecular mechanisms functioning in the female livers will reveal new therapeutic targets. Earlier studies have reported the role of sex hormones (*Liu et al., 2017*; *Yu et al., 2003*; *Ma et al., 2008*; *McGlynn et al., 2015*), transcription factors FoxA1/A2 (*Li et al., 2012*), and cytokine Il6 signaling (*Naugler et al., 2007*) in regulating the sex difference in hepatocellular carcinoma (HCC), but the role of metabolic pathways remains poorly understood.

Rewiring of cellular metabolism enables the tumor cells to maintain viability and grow disproportionately (*Pavlova and Thompson, 2016*). We previously showed that the combined deletion of nuclear receptors, Farnesoid X Receptor (FXR, NR1H4), and Small Heterodimer Partner (SHP, NR0B2) resulted in spontaneous liver cancer in the year-old male mice (*Anakk et al., 2013*). In this study, we report that, unlike the males, female $Nr1h4^{-/-}$, $Nr0b2^{-/-}$ ($Fxr^{-/-}$, $Shp^{-/-}$) double knockout (DKO) mice exhibit protection against tumorigenesis and thus mimic the sexual dimorphism in liver cancer incidence observed in clinics. Although 15-month-old individual $Nr1h4$ knockout and individual $Nr0b2$ knockout mice were previously shown to develop liver cancer, unlike the DKO mice, their incidence does not show 100% penetrance nor sex differences (*Kim et al., 2007*; *Zhang et al., 2008*; *Yang et al., 2007*).

Mutations and reduction in $Nr1h4$, and $Nr0b2$ transcript levels have been noted in cholestasis (reduced bile flow and subsequent increase in hepatic and serum bile acids [(BA)]), fatty liver disease, and liver cancer (*Gomez-Ospina et al., 2016*; *Van Mil et al., 2007*; *Nishigori et al., 2001*; *Kong et al., 2009*; *Wolfe et al., 2011*; *Park et al., 2010*; *He et al., 2008*). Moreover, individuals with chronic cholestasis exhibit an increased risk for HCC (*Eaton et al., 2013*; *Strautnieks et al., 2008*; *van Wessel et al., 2020*). Typically, BA levels are tightly controlled via receptor signaling, including NR1H4 and NR0B2 (*Parks et al., 1999*; *Russell, 2003*; *Thomas et al., 2008*; *Wang et al., 1999*; *Wang et al., 2002*). Consistently, combined loss of $Nr1h4$ and $Nr0b2$ in mice results in juvenile onset cholestasis that progresses to HCC (*Anakk et al., 2011*). We and others have shown that excessive accumulation and dysregulation of BA homeostasis are directly linked with liver cancer risk (*Anakk et al., 2013*; *van Wessel et al., 2020*; *Bernstein et al., 2009*; *Sun et al., 2016*; *Xie et al., 2016*). However, whether BAs are contributing factors to the sex differences seen in HCC prevalence has not been evaluated.

Therefore, we performed transcriptomic analysis to identify distinct gene profiles from both sexes of control and DKO mice. Then, using five separate human clinical HCC cohorts, we tested the clinical utility of the identified gene signatures from our mouse model. Next, we investigated the role of endogenous estrogen signaling in the DKO mice by performing ovariectomy. We measured hepatic, serum, urine, and fecal BAs from male and female mice to understand their homeostasis. Finally, we manipulated the circulating BA levels in the DKO mice either with a chemical challenge or BA binding resins and examined its consequence on hepatic tumorigenesis. Overall, our data uncover that the differential BA homeostasis between the two sexes can orchestrate the observed gender differences in HCC burden in clinics.

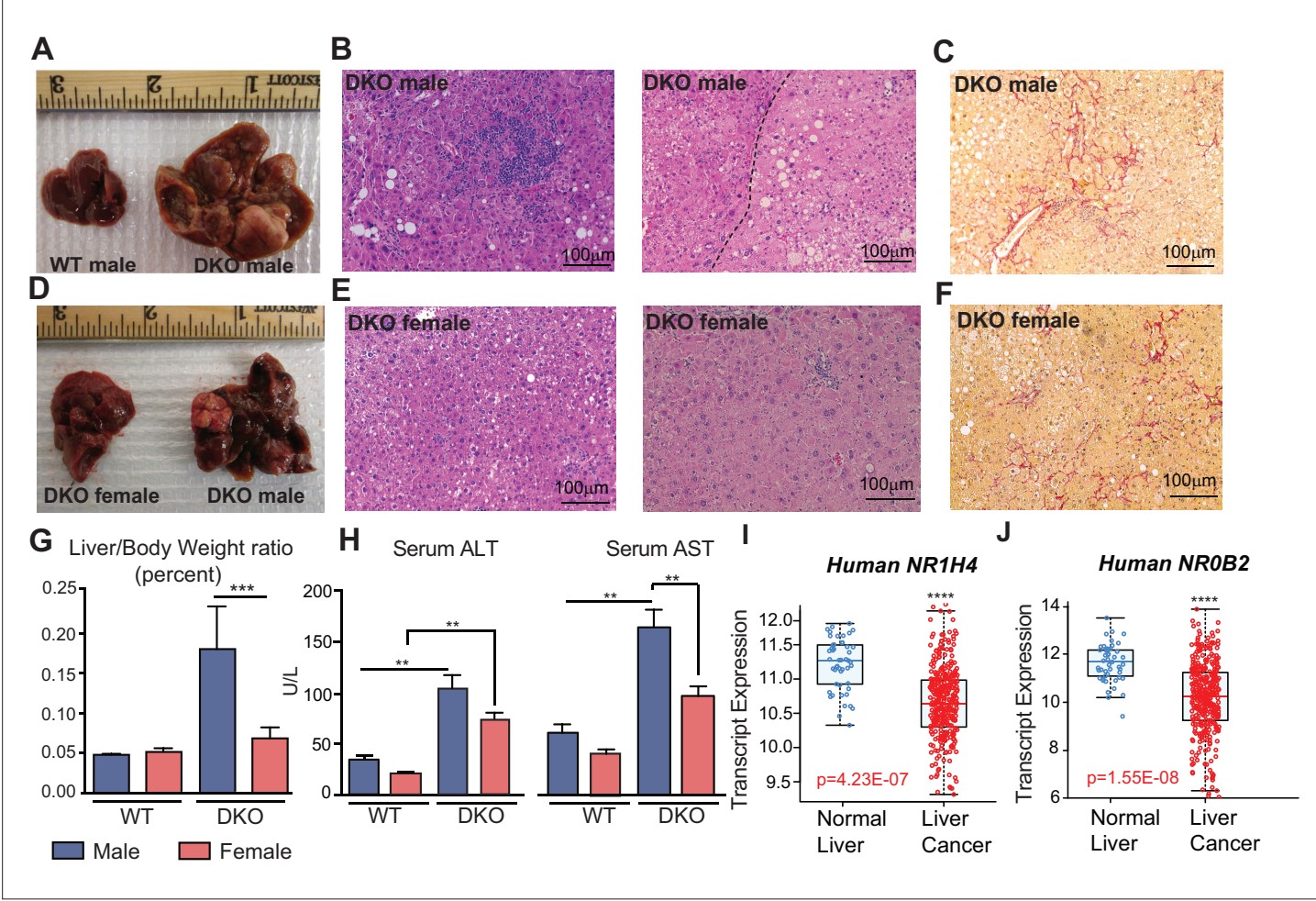

**Figure 1.** *Fxr/Shp* double knockout (DKO) mouse model recapitulates sex difference observed in HCC incidence. (**A**) One-year-old DKO male mice developed hepatocellular carcinoma, which was not observed in age-matched wild type (WT) and DKO female mice (**D**). (**B and E**) Representative H&E stained liver sections from a (**B**) DKO male and (**E**) DKO female. Inflammation and injury are evident at 1 year, and the dotted line (**B**) separates the HCC with large nuclei on the lower right. (**C and F**). Sirius red staining shows increased collagen in a perisinusoidal distribution, which is greater in the DKO males. (**G**) The liver-to-body weight ratio was significantly higher in DKO males. (**H**) Compared to WT animals, serum markers of liver injury AST and ALT were higher in DKO mice. (**I–J**) Analysis of five different HCC clinical cohorts (n=1000) reveals a reduction in *NR1H4* and *NR0B2* transcript levels in patients with liver tumors. n=5–10 mice /group; mean ± SEM; *p<0.01, **p<0.001 compared to genotype or gender controls. One-way ANOVA with Bonferroni post hoc analysis was performed.

The online version of this article includes the following figure supplement(s) for figure 1:

**Figure supplement 1.** DKO female mice exhibit reduced liver injury.

# Results

## DKO mice phenocopy clinical features of HCC

Here, we report that DKO mice exhibit the sexually dimorphic incidence of HCC observed in the clinic. Despite the loss of BA homeostatic machinery, one-year-old DKO female mice did not develop liver tumors. On the contrary, DKO male livers revealed HCC and well-defined adenomas. But both sexes of a year -old DKO mice displayed fat accumulation, and fibrotic sirius red staining (*Figure 1A–F*). At six months of age, DKO female livers were smaller and displayed reduced sirius red staining, indicative of lower hepatic fibrosis compared to males (*Figure 1—figure supplement 1*). The difference in tumor burden was reflected in the gross liver to body weight ratio, which was significantly higher in DKO males than in DKO females (*Figure 1G*). Serum ALT and AST were elevated in the DKO animals compared to WT, consistent with the cholestatic phenotype of these mice. However, these markers were higher in DKO males, corroborating with advanced disease (*Figure 1H*). More importantly,

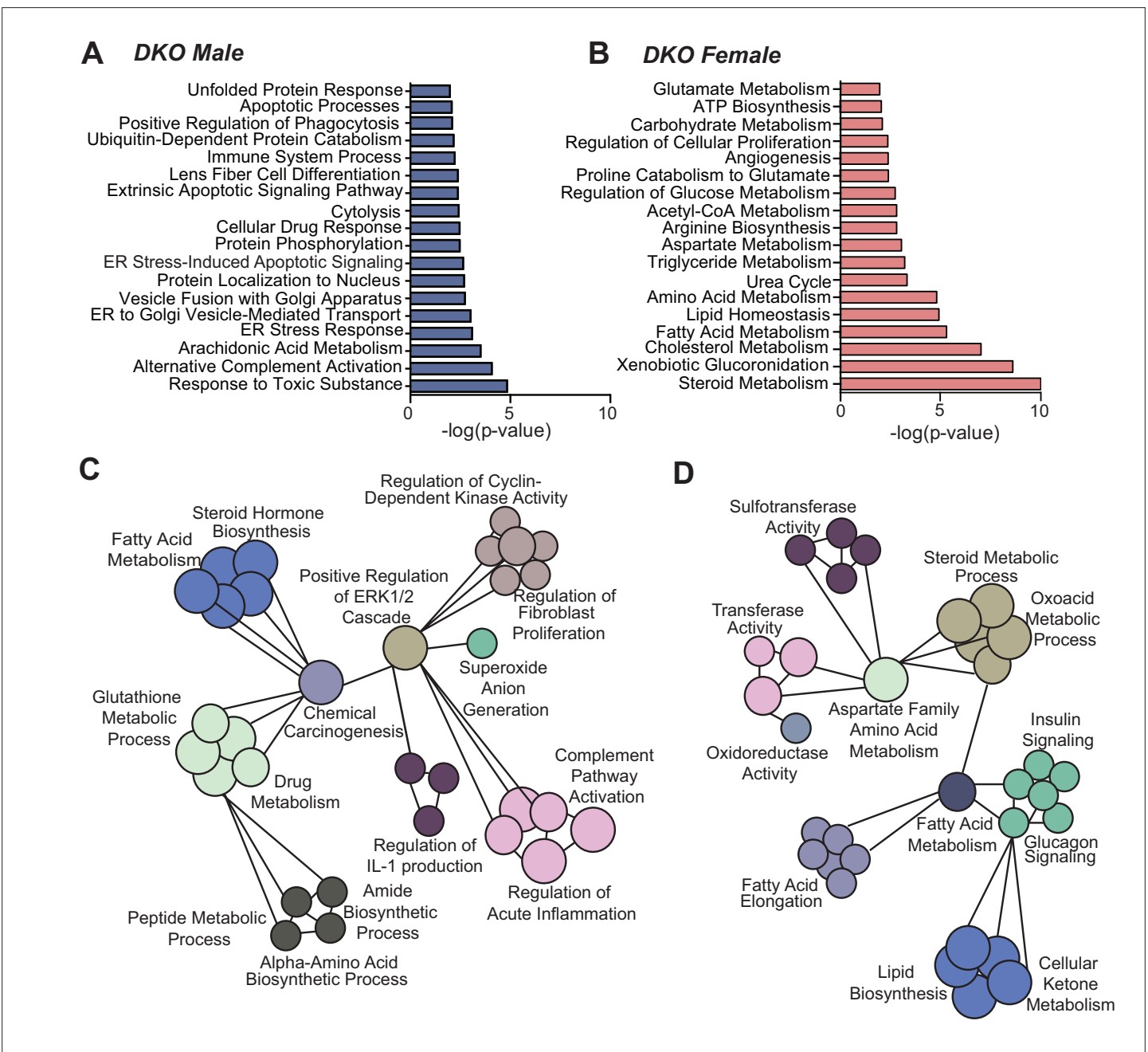

**Figure 2.** Transcriptome analysis reveals striking sex differences in hepatic metabolism. Microarray was performed on liver tissue from WT and DKO mice of both sexes (n=6/group). (**A–B**) GO categories were determined using genes with >1.3 fold change in expression between DKO males and females. Enrichment of overlapping GO categories between males and females was determined by comparing – log p-values for each term. (**C**) GO categories unique to the set of genes upregulated >1.3 -fold in DKO males and (**D**) DKO females.

The online version of this article includes the following source data for figure 2:

**Source data 1.** List of genes in different DKO gene signature categories used for analysis.

**Source data 2.** Transcription motifs enriched in DKO male livers compared to DKO females.

**Source data 3.** Transcription motifs enriched in DKO female livers compared to DKO males.

*NR1H4* and *NR0B2* transcript expression was reduced in liver cancer patients (*Figure 1I–J*), underscoring the clinical relevance of this DKO mouse model.

## Sex-specific metabolic programs regulate liver tumorigenesis

To identify transcriptional mechanisms that can contribute towards the sex differences in the incidence of hepatic tumorigenesis, we analyzed one-year-old male and female livers. DKO males and females displayed striking differences in hepatic gene expression profile (*Figure 2*, GEO GSE151524, and *Figure 2—source data 1*), with DKO males showing enrichment of endoplasmic reticulum stress, unfolded protein response, and immune function (*Figure 2A–B*). Additionally, network analysis with ClueGO (*Bindea et al., 2009*) revealed interactions between drug metabolism, inflammation, ERK signaling, and steroid metabolism in DKO males (*Figure 2C*). On the contrary, DKO females displayed pathway enrichment of steroid metabolism and clustering of lipid, glucose, and amino acid metabolism, along with increased sulfotransferase activity (*Figure 2D*). Next, we parsed the sex-specific upregulated gene sets to identify unique transcription factor motifs. Overrepresented motifs in DKO males, including AR, FOXA1, FOXA2, NRF2, and PPARγ (*Li et al., 2012*; *Koga et al., 2001*; *Ma et al., 2014*; *Schaefer et al., 2005*; *Zhang et al., 2015*), correlated with tumor-promoting functions (*Figure 2—source data 2*). In contrast, in DKO females, FOXO1, E2F, and ERα (*Figure 2—source data 3*) were dominant motifs and are associated with regulating metabolic function during liver carcinogenesis (*Dong et al., 2017*).

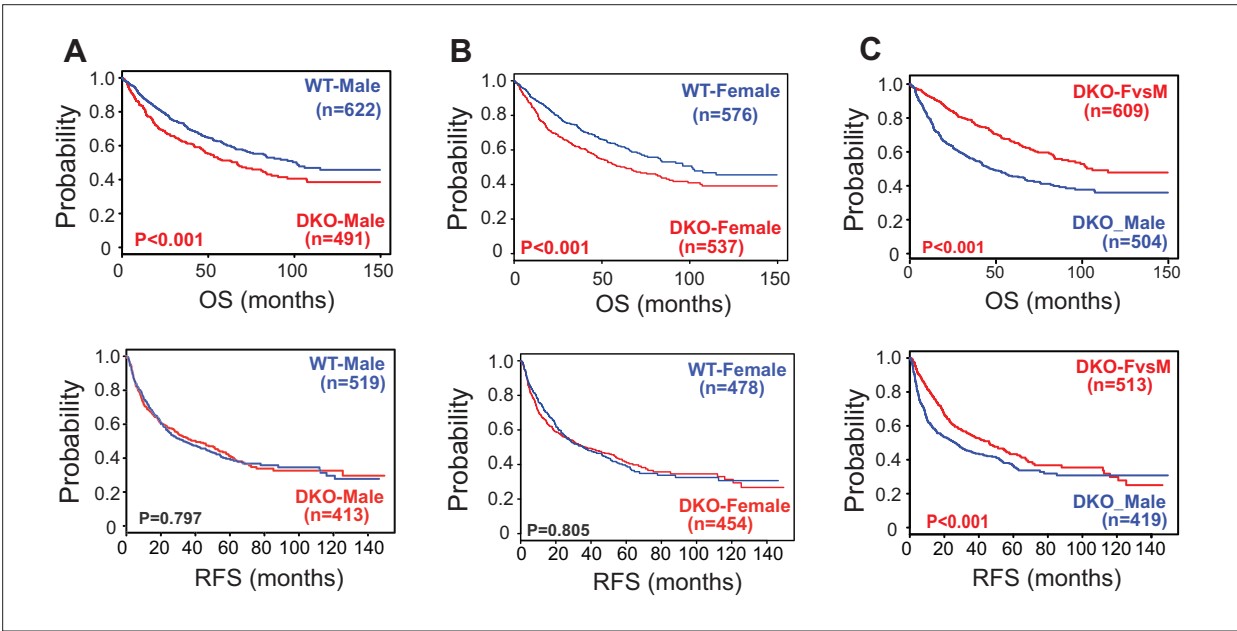

**Figure 3.** Correlation of gene signatures obtained from WT and DKO mouse model with the survival data of HCC patients. The Kaplan Meier Survival graphs were generated based on WT and DKO transcriptome changes using five different HCC clinical cohorts. (**A–C**) Analysis of OS (Overall Survival) and RFS (Recurrence Survival) in patients using the gene signatures representative of either (**A**) male WT or male DKO, (**B**) female WT or female DKO, and (**C**) unique changes observed in female DKO mice but not in male DKO mice.

The online version of this article includes the following source data and figure supplement(s) for figure 3:

**Source data 1.** Table of different HCC gene expression data sets used for analysis.

**Figure supplement 1.** Schematic of the pipeline used to obtain gene signatures to predict outcomes in HCC patient cohorts.

**Figure supplement 2.** Association of DKO-derived gene signatures with clinical stages of HCC.

**Figure supplement 3.** Expression of the urea cycle genes in human liver cancer.

**Figure supplement 4.** DKO female livers exhibit increased expression of urea cycle genes, and they correlate with better patient survival.

## The transcriptomic signature of the DKO mice correlates with poor overall survival in the clinical datasets

To investigate the clinical relevance of the DKO mouse model, we analyzed the WT and DKO murine transcriptomic signatures in a sex-specific manner and compared these to five separate clinical HCC datasets (*Figure 3—source data 1*, *Figure 3—figure supplement 1*). The patient data were sorted based on similarity to one-year-old DKO gene signatures using class prediction (*Figure 3 – Figure 3—figure supplement 1*). Computational prediction scores (BCCP: 1 represents complete match and 0 represents no match) using the patient samples revealed that the DKO_Combined, DKO male signature (DKO_M), and DKO female signature (DKO_F) matched with the later stages (≥2) of liver cancer based on the CLIP (Cancer of the Liver Italian Program) score (*Figure 3—figure supplement 2A–C*), whereas DKO female-specific signature distinct from DKO males (DKO_Fvs.M) matched well with earlier tumor stages (CLIP score 0 or TNM stage 1;) (*Figure 3—figure supplements 2D and 3H*). In contrast, DKO-Combined, DKO_M, and DKO_F matched well with advanced TNM stage 4 (*Figure 3—figure supplement 2E–G*). Fewer individual dots seen in higher CLIP and TNM stages is due to the small number of patients in those categories, and with most of them displaying similar Bayesian prediction probabilities, they cluster around median and overlap.

Of the 1100 patient data, we found approximately (~45%) showed a transcriptomic signature similar to that of either DKO male or DKO female, which corresponded to lower overall survival (OS), but not recurrence-free survival (RFS). WT gene signatures were used as controls (*Figure 3A–B*). Although DKO female mice do not develop liver cancer, it is pertinent to note that they lack *Nr1h4* and *Nr0b2* gene expression, display chronic cholestasis similar to their male counterparts, and hence the global gene changes associate with poor OS. On the contrary, when we focused on the gene signature that was distinctly changed only in the DKO female livers, not the DKO males, we found that patients (~54.71%) who displayed this subset of gene signature had better OS as well as RFS (*Figure 3C*). These findings reveal clinical translational potential for data generated from the DKO mouse model. Moreover, by focusing on specific transcript changes in the DKO female livers, we uncovered a subset of metabolic genes that correspond to better survival and might be responsible for their protection against cancer.

We initially examined the pathways pertaining to amino acid metabolism and ureagenesis since individuals with mutations in the urea cycle disorder have an increased risk of developing liver cancer (*Hashash et al., 2012*; *Koo et al., 2017*; *Rabinovich et al., 2015*; *Wilson et al., 2012*). Consistent with this, analysis of the TGCA-LIHC clinical dataset revealed a broad downregulation of genes encoding the entire urea cycle, including carbamoyl phosphate synthetase (*Cps1*), ornithine transcarbamylase (*Otc*), argininosuccinate synthetase (*Ass1*), argininosuccinate lyase (*Asl*), and arginase (*Arg1*) in both sexes upon liver tumorigenesis (*Figure 3—figure supplement 3*). In contrast, these genes were all upregulated in DKO female livers (*Figure 3—figure supplement 4A*), which correlated well with the protection afforded to the DKO female livers as loss-of-function mutations in these genes are linked to HCC (*Wilson et al., 2012*; *Lam et al., 2009*; *Wu et al., 2021*). Additionally, our analysis showed that patients with increased expression of urea cycle genes (DKO-UreaCycle) exhibited a better clinical outcome (*Figure 3—figure supplement 4B*).

## Estrogen signaling controls amino acid and bile acid metabolism in the liver

Since estrogen signaling was previously shown to regulate amino acid metabolism (*Della Torre et al., 2016*), we examined its role in controlling the expression of urea cycle genes in the DKO female livers. To do this, we ovariectomized (OVX) DKO mice and found that, indeed, the hepatic expression of all these genes, *Cps1, Asl1, Ass, Otc,* and *Arg1* were significantly blunted in the absence of endogenous estrogen signal (*Figure 4—figure supplement 1A*). But when we measured the urea cycle metabolites, we did not find any significant change in the intermediate nor urea production except for a decrease in ornithine levels (*Figure 4—figure supplement 1B*), in DKO females compared to the DKO males. We reason that static measurements may not reflect the flux into the urea cycle.

Besides amino acid metabolism, estrogen signaling has been shown to affect BA homeostasis and cause cholestasis (*Bossard et al., 1993*; *Stieger et al., 2000*; *Yamamoto et al., 2006*). So, we anticipated that OVX would lower BA levels in DKO female mice. Instead, we found that OVX led to liver cancer development in otherwise resistant year-old DKO female mice (*Figure 4A–B*). Moreover,

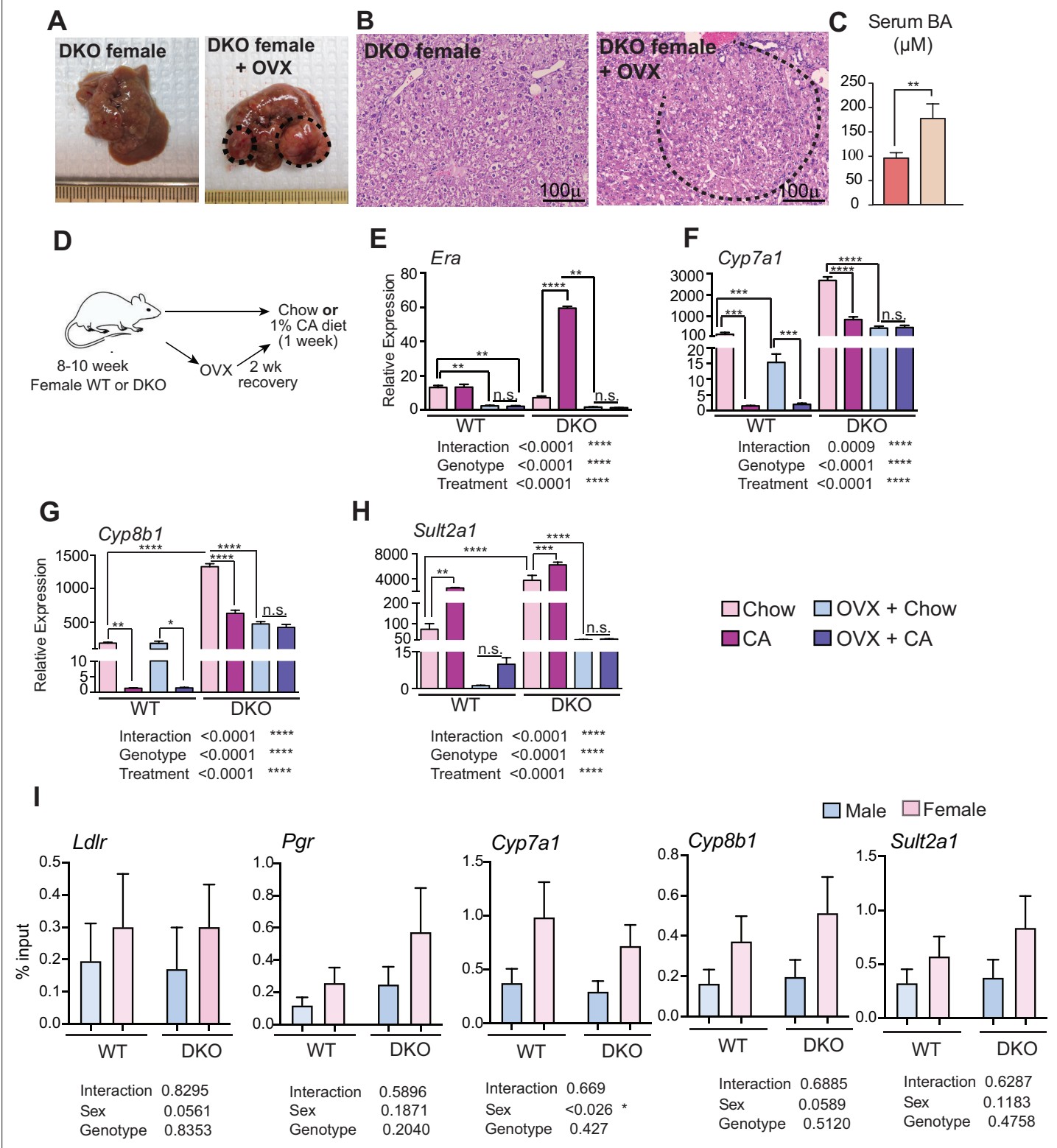

**Figure 4.** Estrogen signaling protects against liver tumorigenesis and may regulate BA synthesis in DKO female mice. (**A–B**) Ovariectomized female DKO mice were aged to a year and examined for liver tumorigenesis, where a dotted line demarcates the tumor margin. (**C**) Serum total bile acid concentrations. (**D**) Experimental design of chow and 1% cholic acid (CA) diet for 1 week with or without (OVX). (**E**) Expression of hepatic *Era* was induced with a CA diet in DKO female mice and reduced in both WT and DKO females following ovariectomy. (**F**) CA-mediated suppression of *Cyp7a1* and (**G**) *Cyp8b1* in WT and DKO females was lost in DKO females after OVX. (**H**) *Sult2a1* has greater baseline expression in DKO mice, induced to a

*Figure 4 continued on next page*

*Figure 4 continued*

lesser extent upon CA challenge compared to WT animals (n=4–5/group). (**I**) ChIP-PCR was performed in WT and DKO male and female livers to test ERa recruitment to BA synthesis and metabolism genes, *Cyp7a1*, *Cyp8b1*, and *Sult2a1*. Rabbit IgG was used as a control for the ChIP-PCR experiments. Mean ± SEM; Two-way ANOVA with Bonferroni post hoc analysis was performed.*$p<0.05$, **$p<0.01$, ***$p<0.001$, ****$p<0.0001$ compared to controls.

The online version of this article includes the following figure supplement(s) for figure 4:

**Figure supplement 1.** Loss of endogenous estrogen lowers the expression of urea cycle transcripts in DKO females.

**Figure supplement 2.** Estrogen receptor signaling positively correlates with better survival in HCC clinical samples.

their serum BA levels doubled (*Figure 4C*), consistent with the tumorigenic role of BAs. Also, analysis of TCGA-LIHC clinical data revealed significant downregulation of *ERα* gene expression in liver tumors (*Figure 4—figure supplement 2A*). In addition, estrogen signaling gene signature obtained from the DKO livers correlated with better overall- and recurrence-free survival (*Figure 4—figure supplement 2B*). Importantly, these results corroborated well with clinical observations that post-menopausal women exhibit higher susceptibility to developing HCC, which can be mitigated upon hormone replacement therapy (*Hassan et al., 2017*; *Wang et al., 2022*).

To overcome the confounding effects of ageing and cancer, we examined young WT and DKO female mice with and without OVX. Additionally, we challenged these mice with BA excess (*Figure 4D*). As expected, OVX resulted in the reduction of basal hepatic *Era* gene expression in both WT and DKO mice (*Figure 4E*). In the DKO mice, which display high basal levels of BA synthesis and sulphation genes, we found dramatic induction of *Era* gene upon BA treatment (*Figure 4E*). Importantly, the rise in *Era* gene coincided with reduced expression of *Cyp7a1*, *Cyp8b1*, and increased levels of *Sult2a1*, a sulphotransferase known to sulphate estrogen and BAs. OVX in WT mice led to lower basal levels of *Cyp7a1* and *Sult2a1* but not *Cyp8b1*, whereas all three genes were significantly reduced in the DKO +OVX livers (*Figure 4G-H*). Unlike the OVX WT, which maintained CA-mediated suppression of BA synthetic genes, consistent with intact NR1H4 signaling, DKO +OVX mice did not alter their expression (*Figure 4F–G*). These data indicate a role for estrogen signaling in regulating BA homeostasis in the DKO livers.

We next examined if the recruitment of ERa to BA synthesis genes exhibited any sex difference in WT and DKO livers by ChIP-PCR. We find that ERa was preferentially recruited to *Cyp7a1* in a sex-specific manner (*Figure 4I*). This is exciting because CYP7A1 is the rate-limiting enzyme in classical BA synthesis and is responsible for generating a majority (~75%) of BAs. *Cyp8b1* showed a similar trend but not *Sult2a1*. Also, we did not find any sex-specific patterns in ERa occupancy in *Ldlr* and *Pgr* genes, which were used as positive controls for ERa ChIP assays (*Figure 4I*). These data, along with increased BAs upon OVX, suggest ERa signaling is pertinent to control BA synthesis, especially in the absence of NR1H4, as seen in the CA-fed sham DKO mice.

## DKO mice display sexual dimorphism in BA homeostasis

WT mice do not show overt changes in serum BAs between the two sexes. Although serum BA concentration in DKO females was higher than in WT females; however, compared to genetically identical DKO males, these female mice displayed lower primary and secondary serum BAs (*Figure 5—source data 1*). This was intriguing.

So, we analyzed the expression of genes involved in BA synthesis, transport, and metabolism in both sexes of DKO mice. Consistent with *Nr1h4* and *Nr0b2* deletion that results in the loss of negative feedback on BA biosynthesis, both sexes of DKO mice have significantly higher expression of *Cyp7a1* and *Cyp8b1* genes that are involved in classical BA synthesis (*Figure 5A*). The male dominant expression of *Cyp7b1* in the WT is lost in the DKO mice. On the other hand, *Cyp27a1*, which initiates alternative BA synthesis, was increased in a female-specific manner (*Figure 5A*).

Next, we examined BA transport. We found that hepatic transcript levels of the key BA efflux pump, bile salt export pump, *Abcb11 (Bsep)*, were reduced in both sexes of DKO mice, consistent with loss of *Nr1h4* (24) (*Figure 5B*). In contrast, the expression of canalicular efflux transporters, *Abcb1 (Mdr1)* and *Abcc2 (Mrp2)*, was unchanged (*Figure 5B*). Also, the BA uptake transporter, sodium taurocholate co-transporting polypeptide, *Slc10a1 (Ntcp)* showed lower transcript levels in females (*Figure 5B*), which is in line with previous findings that estradiol can downregulate *Slc10a1* expression (*Simon et al., 2004*).

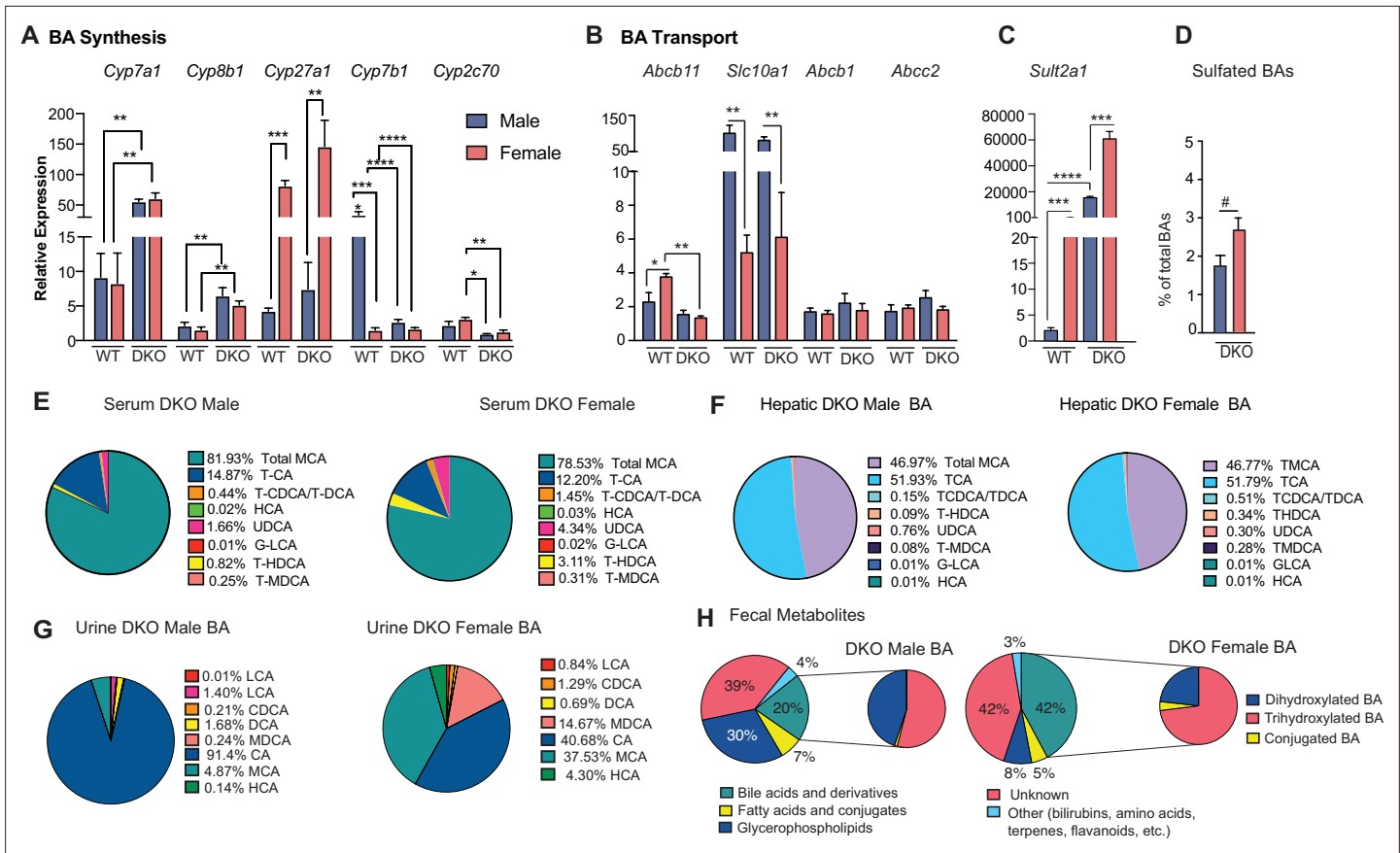

**Figure 5.** BA composition and metabolism are differentially regulated between the sexes of DKO mice. (**A**) Hepatic mRNA expression of classical BA synthetic enzymes was elevated in 6-month-old DKO compared to WT mice. While the alternative BA synthesis encoding gene, *Cyp27a1*, was increased in females only. (**B**) Expression of hepatic BA transporters and (**C**) BA sulfotransferase in WT and DKO mice. (**D**) Percentages of sulfated BAs in DKO male and female serum (one-tailed t-test, #*P*p<0.05). (**E–F**) BA composition is slightly varied in serum, whereas it remains unchanged in the liver between DKO males and females. (**G–H**) BA composition in the urine was variable between the sexes, and BAs constitute a higher proportion of fecal metabolites in the year old DKO females compared to males (n=5–7/group). Mean ± SEM; *p<0.05, **p<0.01, ***p<0.001, ****p<0.0001 compared to genotype or sex-specific controls. One-way ANOVA with Bonferroni post hoc analysis was performed.

The online version of this article includes the following source data for figure 5:

**Source data 1.** Serum BA composition in DKO mice.

**Source data 2.** Urine BA composition in DKO mice.

We then investigated the transcript expression of *Sult2a1*, which contributes to BA sulfation—a modification that can reduce enterohepatic recirculation (*De Witt and Lack, 1980*). As expected, hepatic *Sult2a1* expression was predominant in females irrespective of the genotype (*Figure 5C*; *Alnouti and Klaassen, 2011*). Sulphated BAs are excreted in urine to eliminate excess BA during cholestasis (*Heuman, 1989*; *Stiehl et al., 1975*). Total urine BA levels were higher in DKO males, reflecting a larger circulating BA pool than in DKO females (*Figure 5—source data 2*). However, DKO female mice exhibited higher percentages of sulphated BAs (*Figure 5D*), which corroborates with high *Sult2a1* expression in females.

BA compositional analysis was performed in the serum, hepatic, urine, and feces of DKO males and females (*Figure 5E–H*). Both sexes of DKO mice showed abundant muricholates in the serum (*Figure 5E*), but there were modest differences in the composition, indicating slightly hydrophilic BAs in the DKO females. Moreover, hepatic BA composition was indifferent between the two sexes (*Figure 5F*). These results indicated that rather than synthesis or transport, excretion may be different between DKO males and females. Notably, we found that both urine and fecal levels and composition between male and female DKO mice were distinct (*Figure 5G–H*). As urinary BA excretion alone cannot explain the 50% decrease in circulating BAs in DKO females, we performed untargeted

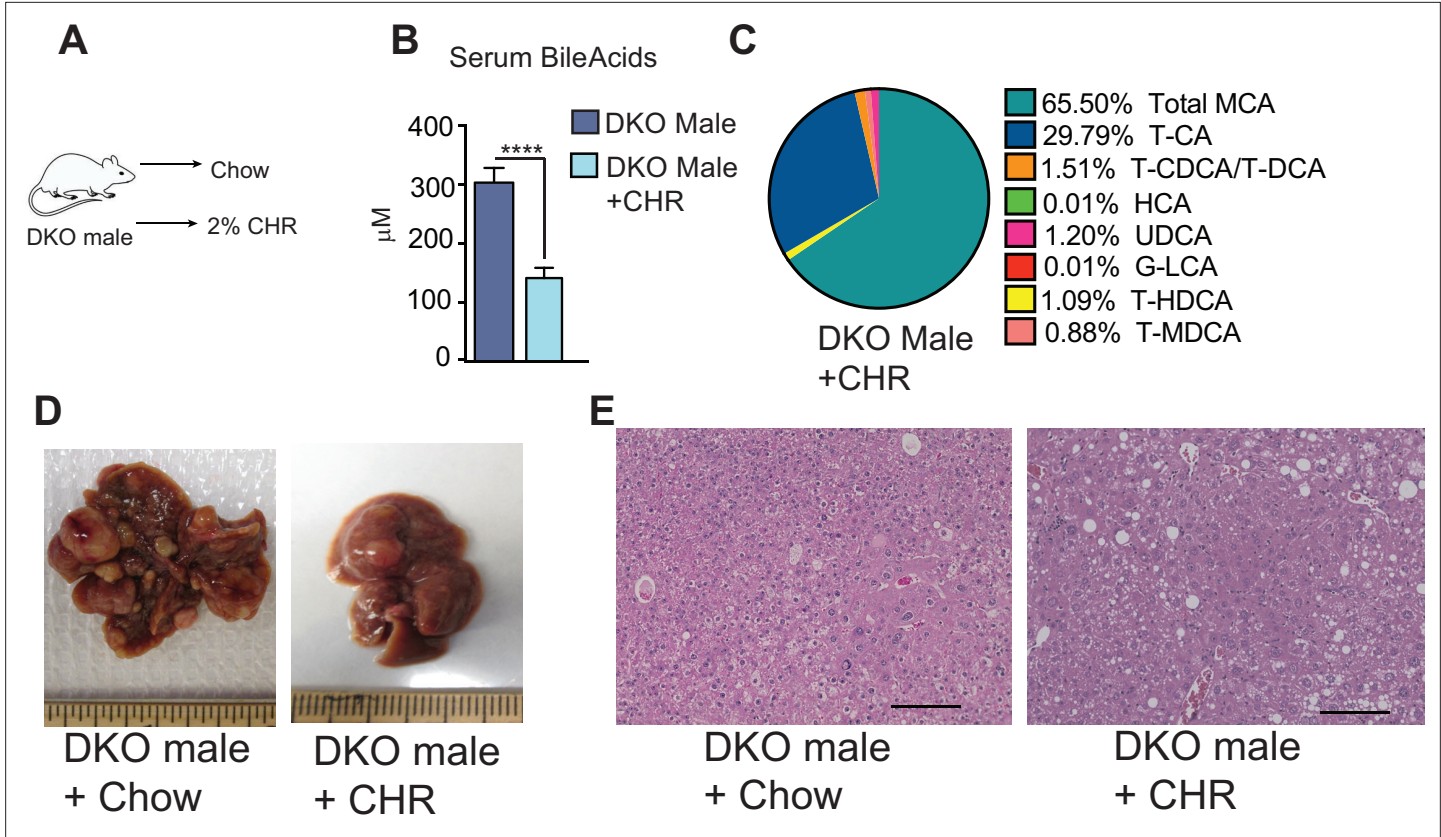

**Figure 6.** Treating with BA-binding resin reduces the tumor burden in DKO male mice. (**A**) DKO male mice were fed a 2% cholestyramine (CHR)-enriched diet for 3 months until 1 year of age. (**B–C**) Serum BA levels and composition upon feeding DKO male mice a CHR-enriched diet. (**D**) CHR dramatically reduced the HCC burden in DKO males. (**E**) Histological analysis shows HCC, bland tumor cells, and enlarged nuclei with irregular membranes in DKO male mice. CHR treatment results in smaller and fewer nodules but increases steatosis. (n=6–7/group). Mean ± SEM; ****p<0.0001 compared to DKO controls.

The online version of this article includes the following source data and figure supplement(s) for :

**Source data 1.** Hepatic and serum BA profiles after different diet regimens (Chow, DDC, or CHR) in WT and DKO mice.

**Figure supplement 1.** DKO females challenged with the DDC-enriched diet developed liver tumors.

**Figure supplement 2.** BA composition and expression of BA synthesis genes in DKO mice after different treatments.

metabolomics using the fecal samples. BAs accounted for 20% of the fecal samples in the males, whereas in the females, it was double the amount indicative of twice the amount being excreted in DKO females. These results indicate that female DKOs may be protected against detrimental tumor-promoting BA signaling due to their higher BA excretion.

## Increasing fecal BA excretion is sufficient to reduce liver cancer risk

Finally, to test this, we promoted fecal BA excretion in DKO males by using cholestyramine (CHR), a resin that binds BAs. We fed nine-month-old DKO male mice with a 2% CHR-containing diet since, by this age, tumor nodules have already developed. The CHR diet was continued until one year of age, mimicking a therapeutic intervention strategy (*Figure 6A*). As expected, the CHR diet in DKO males led to a 50% reduction in circulating BA levels and altered BA composition (*Figure 6B-C*; *Figure 6—source data 1*). DKO males fed chow exhibited severe hepatic tumorigenesis, whereas CHR-fed DKO males had a drastically lower tumor burden with only small liver nodules and were protected from developing aggressive carcinomas. Histological analysis revealed that CHR treatment lowered the number of nodules and dysplastic changes but increased steatosis in DKO males (*Figure 6D*). Conversely, increasing circulating BAs by causing biliary injury with 3,5-diethoxycarbonyl

–1,4-dihydrocollidine (DDC) in DKO females resulted in the development of large liver tumors in DKO females (*Figure 6—figure supplement 1*).

Unbiased correlation analysis of hepatic and serum BA composition between the two cohorts of DKO mice revealed that the BA profiles of CHR-fed DKO male mice clustered with DKO females, whereas DDC-fed DKO female mice clustered with DKO males (*Figure 6—figure supplement 2*).

Overall, these findings demonstrate that modulating circulating BAs is sufficient to change the liver cancer outcome, wherein lowering their levels leads to subsequent protection and vice versa.

## Discussion

Here, we demonstrate that the sex differences in BA homeostasis can contributes to the sexual disparity noted in HCC risk. Importantly, elevated BA concentrations are reported in patients with HCC (*Anakk et al., 2013*; *van Wessel et al., 2020*; *Bernstein et al., 2009*; *Sun et al., 2016*; *Xie et al., 2016*; *Cameron et al., 1982*). Using a genetic mouse model of excess BAs that develop spontaneous HCC, we uncovered distinct transcriptional control of metabolism between the two sexes. Both *NR1H4* and *NR0B2* transcript levels were downregulated in HCC patients. Moreover, differential gene expression, specifically of the DKO female, correlated well with better survival, highlighting the translational relevance of our model. Thus, these gene signatures could be utilized as a potential prognostic marker for HCC progression and survival.

Both BA homeostasis and amino acid metabolism were altered between the two sexes. Of note, genes controlling ureagenesis were higher in the DKO females, and consistent with previous findings, we were able to recapitulate estrogen-mediated regulation of some of these genes signaling (*Della Torre et al., 2016*). In line with these findings, patients with urea cycle enzyme deficiencies have a 200 x higher incidence of HCC, highlighting the importance of amino acid metabolism in hepatic tumorigenesis (*Hashash et al., 2012*; *Koo et al., 2017*; *Rabinovich et al., 2015*; *Wilson et al., 2012*). Also, BAs have been shown to promote amino acid catabolic machinery (*Massafra et al., 2017*), which indicates that BAs may be a central node in liver cancer. Intriguingly, hepatic urea analysis did not reveal any difference between the DKO male and female mice. A caveat being we measured a steady-state urea levels rather than the flux of this pathway.

We examined and found estrogen signaling can regulate the expression of BA synthesis and sulphation genes. DKO female mice challenged with the CA diet showed a robust increase in hepatic Era transcript, which coincided with the suppression in BA synthesis in the absence of *Nr1h4* and *Nr0b2*. Consistently higher recruitment of ER to the classical BA synthetic gene, *Cyp7a1*, was noted in female livers. Intriguingly, DKO OVX mice with blunted Erα gene expression exhibited a lower transcript level of *Cyp7a1 and Cyp8b1* and lost the CA-mediated suppression. These findings suggest Era expression is linked to both positive and negative regulation of BA synthesis genes. But we do not know how ER elicits these differential transcriptional effects on BA synthesis.

Nonetheless, we confirmed the known sex differences in BA synthesis, such as a female-dominant *Cyp27a1* expression and male-dominant *Cyp7b1* pattern in WT mice. Loss of *Nr1h4* and *Nr0b2* altered the expression of many genes irrespective of sex. For instance, *Cyp2c70* expression was reduced in *both sexes*, and the male dominance of *Cyp7b1* was lost in the DKO mice. Of note, OVX of DKO females increased the serum BA levels and lost their protection against the development of liver tumorigenesis. This finding fully recapitulates the clinical data, wherein post-menopausal women are equally prone to HCC incidence as males.

BA analysis shows that DKO female mice have a hydrophilic composition and excrete BA proportions. So, we tested and demonstrated the potent therapeutic utility of reducing BA levels in serum using a generic FDA-approved BA binding resin, Cholestyramine (CHR), in dramatically reducing the tumor burden. This study highlights that lowering enterohepatic recirculation is a beneficial strategy in modulating liver cancer. Though *Cyp7a1* expression is reported to be induced in CHR-fed mice (*Schwarz et al., 2001*), long-term CHR feeding in DKO mice lowered *Cyp7a1* expression but induced *Cyp8b1* transcripts (*Figure 6—figure supplement 2*). Conversely, DDC-fed DKO females that develop hepatic tumors show a corresponding decrease in *Cyp8b1* transcript (*Figure 6—figure supplement 2*). Also, patients with HCC exhibit a reduction in *Cyp8b1* expression (*Hoenerhoff et al., 2011*; *Grinberg et al., 2014*; *Wang et al., 2019*), which promotes a more hydrophilic ratio of BA composition.

Although species differences in BAs between mice and humans are a limitation, several fundamental understandings have been gained from mouse experiments. While this study demonstrates

BAs promote HCC progression, it does not investigate or provide evidence if BAs are sufficient for HCC initiation. Another caveat is that the DKO mouse model mimics the progression of cholestasis to HCC and not all the etiologies, so the observed sex differences in circulating BAs may be limited to these subsets of HCC. Nevertheless, elevated BA concentrations are seen in various liver disease conditions and inherited disorders of cholestasis predispose to HCC onset. More recently, clinical studies support the utilization of BAs as prognostic markers (*Cui et al., 2023*; *Huang et al., 2020*; *Thomas et al., 2021*).

Our findings demonstrate that female cholestatic mice exhibit increased excretion and lower serum BAs than males. However, hepatic BAs were not different between the sexes. These data highlight that circulating BAs contribute towards sex differences seen in HCC incidence. Accordingly, we show that lowering enterohepatic BA recirculation is beneficial in our model. Our results align with previous findings that had implicated intestinal NR1H4 signaling as being crucial rather than hepatic NR1H4 to prevent liver tumorigenesis (*Degirolamo et al., 2015*). Taken together, these results reveal that drugs inhibiting intestinal reabsorption of BAs (Asbt inhibitor, IBAT inhibitor) that are on clinical trials for NASH and cholestasis can be evaluated as potential therapeutics to combat HCC.

## Materials and methods

**Key resources table**

| Reagent type (species) or resource | Designation | Source or reference | Identifiers | Additional information |
|---|---|---|---|---|
| Gene (*Mus musculus*) | Farnesoid x receptor (FXR) and Small heterodimer partner (SHP) | GenBank | Nr1h4 Nr0b2 | Key regulators of bile acid homeostasis |
| Strain background (*Mus musculus*) | C57BL/6 | In house PMCID:PMC3007143 | FXR/SHP Ddouble knockout (DKO) mice; RRID:MGI:2159769 | Both sexes of DKO model and wild type mice were used for analysis in this paper |
| Chemical compound used in the diet - (0.1%) | 3,5-Di-ethoxycarbonyl-1,4-Dihydrocollidine (DDC) | Purchased from Sigma | Catalog# 137030 | Envigo – (chemical compound was mixed with base diet and pelleted) |
| Chemical compound used in the diet – (2%) | Cholestyramine (CHR) | Purchased from Sigma | Catalog # C4650 | Envigo –(chemical compound was mixed with base diet and pelleted) |
| Other | Hematoxylin & Eosin Sirius Red Staining | Epredia VWR-Avantor | Catalog # 71311, # 7211 Catalog # 10065–176 | Stains used to analyse liver histology |
| Sequence-based reagent | Several genes | This paper | PCR primers | See *Supplementary file 1* |
| Antibody | Anti-rabbit polyclonal ERα-MC10 | sc-542 Santacruz | RRID:AB_631470 | Used for ERα ChIP |
| Commercial assay or kit | Bile acid analysis | Genway Biotech | Total BA concentration | |
| Software | EndNote Prism | Clarivate GraphPad | EndNote Prism | Reference Statistical analysis |

### Experimental design

This study was designed to elucidate the role of bile acids (BAs) in the sexually dimorphic incidence of HCC and assess the therapeutic benefits of reducing circulating BA levels on HCC development. WT mice and $Nr1h4^{-/-}$, $Nr0b2^{-/-}$ ($Fxr^{-/-}$, $Shp^{-/-}$) double knockout (DKO) mice on a C57BL/6 background were bred at the animal facility at the University of Illinois Urbana Champaign to generate sufficient numbers for experimentation. Mice were housed on a standard 12 hour light/dark cycle and fed normal chow and water ad libitum. Male and female WT and DKO mice were sacrificed at three different time points, 8–12 weeks or 6- and 12–13 months after birth. For cancer studies, male and female WT and DKO mice were aged to 1 year. To study estrogen signaling, bilateral ovariectomies were performed on WT and DKO females at 8–10 weeks old, followed by 2 weeks of recovery and were subsequently challenged with a CA-enriched diet for a week. To test the role of estrogen in liver cancer development, another set of ovariectomized DKO mice and their sham controls were aged to a year. To test the effect of liver injury and BA accumulation, 0.1% DDC (3,5- diethoxycarbonyl-1,

4-dihydrocollidine) was fed to 10-month-old DKO female mice for 3 months. In another cohort, 2% CHR (Cholestyramine)-supplemented chow was fed to 9- month-old DKO male mice for a period of 3 months. Urea cycle studies were performed on DKO mice after overnight fasting. Serum and liver tissues were typically collected from all the cohorts. For some of the DKO groups, urine and feces were additionally collected for BA analysis. All studies were carried out as outlined in the Guide for the Care and Use of Laboratory Animals prepared by the National Academy of Sciences and published by the National Institutes of Health (National Institutes of Health publication 86–23, revised 1985). All of the animals were handled according to approved institutional animal care and use committee (IACUC) protocols of the University of Illinois, Urbana Champaign. For biological harvesting, mice were anesthetized and euthanized as described by IACUC. For biological harvesting, mice were anesthetized and euthanized as described by IACUC. Tissue was flash-frozen in liquid nitrogen, and blood was collected and spun down for serum.

## Serum chemistry
Blood was collected by retro-orbital bleeding and centrifuged at 8000 × g xfor 10 minutes to separate serum. Serum ALT and AST were measured using Infinity ALT and Infinity AST kits (Thermo Fisher Scientific). Calorimetric measurement of serum and hepatic BAs was performed with the Total Bile Acid (NBT method) kit (Genway Biotech).

## Bile acid analysis
Serum and urine from DKO male and female mice fed chow were analyzed for the composition of bile acids and their sulfated metabolites at the University of Nebraska Medical Center. Liquid chromatographic-mass spectrometry analysis was performed with a Waters ACQUITY column (Milford, MA). Bile acids and internal standards were prepared in methanol and analyzed. These data are provided as *Figure 5—source data 1 and 2*. Serum from DKO male and female mice fed chow, DKO males fed CHR, and DKO females fed DDC along with hepatic extracts from DKO male and female was analyzed for BA composition at Baylor College of Medicine Metabolomics Core, Houston, Texas. Briefly, liver tissue was weighed and homogenized in 75% ethanol and incubated at 50 °C for 2 hoursr to extract BAs and centrifuged at 6,000 × *g* for 10 minutes. The supernatant was used to determine the BA content. Liquid chromatographic-mass spectrometry analysis was performed with a Waters ACQUITY UPLC BEH C18 column (Milford, MA). Bile acids were detected in negative mode, with L-Zeatine added to each sample as an internal standard. These data are provided as *Figure 6— figure supplement 2*, *Figure 6—source data 1*.

## Metabolite profiling
Liver tissue was weighed and sonicated in 70% methanol, followed by centrifugation. The supernatant was flash-frozen and used for subsequent LC-MS analysis for urea cycle metabolites. Tissue lysate was used for the BCA assay to determine the protein concentration of each sample. All metabolite concentrations were normalized to a protein concentration of the lysate.

## Untargeted metabolomics
Fecal samples were weighed into microcentrifuge tubes and homogenized in 50% MeOH/H2O solution with a 1:10 w/v ratio, for 5 minutes at 5 Hz. The samples were centrifuged at 14,000 rpm for 15 minutes, then a 200 µL aliquot of each supernatant was transferred to a 96-well plate and dried under centrifugal vacuum. The dried extracts were covered and stored at –80 °C until analysis, at which time the samples were resuspended in 200 µL of 50% MeOH/H2O solution with 1 µM sulfadimethoxine as internal standard and diluted three-fold for analysis. Untargeted LC-MS/MS was performed on a Thermo Vanquish UPLC system coupled to a Q-Exactive Orbitrap mass spectrometer (ThermoFisher Scientific, Bremen, Germany). A polar C18 column (Kinetex polar C18, 100x2.1 mm, 2.6 µm particle size, 100 A pore size; Phenomenex, Torrance, CA, USA) was used as the stationary phase, and a high-pressure binary gradient pump was used to deliver the mobile phase, which consisted of solvent A (100% H2O+0.1% formic acid [FA]) and solvent B (100% acetonitrile [ACN] +0.1% FA). The flow rate was set to 0.5 mL/min and the injection volume for each sample was 5 µL. Following injection, samples were eluted with the following gradient: 0–1.0 min, 5% B; 1.0–1.1 min, 25%; 6.0 min, 70%; 7.0 min, 100%; 7.5–8.0 min, 5%. MS data was collected in positive mode and electrospray ionization (ESI)

parameters were set to 53 L/min for sheath gas, 14 L/min for auxiliary gas, 0 L/min for spare gas, and 400 °C for auxiliary gas temperature. The spray voltage was set to 3500 V, the capillary temperature to 320 °C, and the S-Lens radio frequency level to 50 V. MS1 data were collected from 150 to 1500 m/z with a resolution of 35,000 at m/z 200 with one micro scan. The maximum ion injection time was set to 100ms with an automatic gain control (AGC) target of 1.0E6. MS/MS spectra were collected using data-dependent acquisition (DDA), where the top five most abundant ions in the MS1 scan were selected for fragmentation.

Normalized collision energies were increased stepwise from 20, 30, –40. MS2 data were collected with a resolution of 17,500 at m/z 200 with one micro scan and an AGC of 5.0E5. All untargeted LC-MS/MS data are publicly available from the MassIVE data repository under accession number MSV000089715.

MS1 feature detection and MS/MS pairing was performed using MZmine 2.37corr17.7_kai_merge (55, 56). An intensity threshold of 5E4 and 1E3 were set for MS1 and MS2 detection, respectively, with centroid data. MS1 chromatogram construction was performed using the ADAP chromatogram builder, where the minimum group size was set to 5, group intensity threshold was 5E4, minimum highest intensity was 1.5E5, and mass tolerance was 0.005 m/z or 10 ppm. Chromatogram deconvolution was then performed using a local minimum search algorithm with a chromatographic threshold of 80%, a search minimum in retention time (RT) range of 0.2 min, minimum relative height of 1%, minimum absolute threshold height of 1.5E5, minimum ratio for top/edge of 1, and a peak duration of 0.05–2.0 min. Pairing between MS1 and MS2 was performed with a mass tolerance of 0.005 m/z or 10 ppm and RT range of 0.2 min. Isotope peaks were grouped, then features from different samples were aligned using the same mass and RT tolerances; alignment was performed by placing a weight of 75 on m/z and 25 on RT. A peak area feature table was exported as a .csv file and consensus MS/MS spectral data were exported in mgf format. Feature-based molecular networking and MolNetEnhancer workflows were then performed with this data using GNPS (gnps.ucsd.edu). The corresponding jobs can be found: here and here, respectively.

## Histology

Liver samples were collected and fixed in 10% neutral-buffered formalin at 4 °C. Formalin-fixed tissues were then processed and embedded in paraffin. Subsequently, paraffin tissue blocks were sectioned using a microtome at 5 μm thickness and mounted onto charged glass slides. Tissues sections were deparaffinized and stained with hematoxylin & eosin (H&E;) (Thermo Fisher Scientific) and sirius red staining using standard histological protocol.

## RNA extraction and quantitative PCR analysis

Total RNA from the liver was prepared according to the TRIzol (Invitrogen) protocol. cDNA was synthesized using Maxima Reverse Transcriptase (Thermo Fisher Scientific) as per the manufacturer's protocol. q-RTPCR was performed on an Illumina Eco Platform. For qRT-PCR analysis, 50 ng of cDNA was added to each SYBR green-based reaction. qRT-PCR primers are provided in *Supplementary file 1*.

## Microarray

Microarray was performed by Dr. Ju-Seog Lee's laboratory at the MD Anderson Cancer Center. Liver samples from 12-month-old male and female WT and DKO mice were collected and snap-frozen. Total RNA was isolated, labeled, and hybridized to BeadChip Array MouseWG-6 (Illumina). Bead chips were scanned with an Illumina BeadArray Reader. Microarray analysis was performed on the Illumina mouseRefseq-8 Expression platform. Upregulated gene sets were generated from genes with fold change >1.3 (*P*p<0.0001) compared to the control group (i.e. DKO males vs. DKO females). These gene sets were then used for downstream analyses with DAVID Bioinformatics Resources Analysis Software and ClueGO (*Bindea et al., 2009*).

## Transcription factor motif analysis

GeneXplain software was used to identify enriched transcription factor binding sites (TFBS) using the upregulated gene sets generated from the microarray. The analysis included regions from –1000 to –100 bp relative to the transcription start site. TFBS enriched with *P*≤0.01 were included in the tables.

## Extraction of transcriptomic signature

Multiple transcriptomic signatures were extracted from the microarray data of the DKO mouse model (*Supplementary file 1*). DKO_All signature was generated from the comparison between wild type (WT) male and female mice, and DKO_Male and DKO_Female signatures from WT male and female mice, respectively.

DKO_FvsM, DKO_Estrogen, DKO_BA, and DKO_Urea signatures were made from the comparison between DKO male and female mice. Signature genes were selected by T-test and logFC ($P<0.001$ and log2FC $>1$ or $<-1$) using the gene expression dataset after normalization.

## Gene expression data from HCC tumors

Gene expression data from the National Cancer Institute (NCI) cohort were generated in earlier studies (*Lee et al., 2004a*; *Lee et al., 2004b*; *Lee et al., 2006*), and the data are publicly available from the NCBI's GEO database (GSE1898 and GSE4024). Gene expression data from Korea, Samsung, Modena, and Fudan cohorts have been described previously and are available from the NCBI's GEO database (accession numbers, GSE14520, GSE16757, GSE43619, GSE36376, and GSE54236;) (*Kim et al., 2012*; *Park et al., 2016*; *Roessler et al., 2010*; *Sung et al., 2012*; *Villa et al., 2016*). TCGA RNA sequencing data for HCC was downloaded from the University of California, Santa Cruz, Genomics Institute (https://xenabrowser.net/). FPKM-normalized data were log-transformed.

Tumor specimens and clinical data were obtained from HCC patients who had undergone hepatectomy as a primary treatment for HCC at multiple institutes, as described in their original study. Except for the TCGA cohort, patients and tissues were collected based on the availability of high quality of frozen tissues for genomic studies. For the TCGA cohort (*Cancer Genome Atlas Research Network, 2017*), surgical resection of biopsy biospecimens were collected from patients diagnosed with HCC and who had not received prior treatment for their disease (ablation, chemotherapy, or radiotherapy). Institutional review boards at each tissue source site reviewed protocols and consent documentation and approved the submission of cases to TCGA. Hematoxylin and eosin (H&E) stained samples were subjected to independent pathology review to confirm that the tumor specimen was histologically consistent with the allowable HCC. Each case was reviewed independently by at least 3 liver pathologists, with no clinical or molecular information.

## Data analysis of clinical samples

To predict the class similar to the DKO signature in the human HCC cohort, we used a classification algorithm based on Bayesian compound covariate predictor (BCCP). After the integration of the signature matrix and the human HCC dataset, the Bayesian probability for each human HCC sample was calculated by using the class prediction procedure in BRB Arraytools. Before pooling mouse and human gene expression data for analysis, expression data of orthologous genes in both data sets were independently converted to z-scores ($z = (x − mean)/standard deviation$). A Bayesian compound covariate prediction (BCCP) algorithm was used to estimate the probability that a particular human HCC sample would have a given gene expression pattern from mouse tissue (69, 70). Prognostic significance was evaluated rigorously for overall and recurrence-free survival in the human HCC cohort based on the predicted class calculated by the BCCP algorithm using multiple DKO signatures. A total of 5 human HCC transcriptomic cohorts were used in this study (Fudan, Korea, Samsung, TCGA, Modena). All DKO signatures were evaluated in each human HCC cohort and meta-cohort. To identify the gender difference in the human HCC cohort, we did subgroup analysis for gender and age in the meta-cohort. BCCP scores (BCCP probability) were compared in all populations and gender subgroups. The analysis for potential correlation between the class predicted by DKO-signature and staging HCC in terms of TNM, and CLIP classification was performed in the meta-cohort.

## ERα ChIP analysis

ERa-ChIP assay was performed in both sexes of WT and DKO mice. We first analyzed BED files for ERα ChIP-Seq from three independent studies (*Gertz et al., 2013*; *Gordon et al., 2014*; *Palierne et al., 2016*) obtained from Cistrome DB. We visualized the tracks on UCSC genome browser on mouse GRCm38/mm10 assembly to identify potential binding sites on genes that maintained BA homeostasis. Then, primers were designed for those regions to validate ERa binding using ChIP-PCR. Briefly, ERa antibody (MC20, Santa Cruz, # sc-542, RRID:AB_631470) was used to perform chromatin

pulldown from flash-frozen liver tissues with rabbit IgG chromatin pulldown as controls. ChIP DNA was isolated using a QIAGENiagen PCR purification kit, followed by qPCR to examine the recruitment to the region.

## Statistical analysis

All statistical tests were performed using GraphPad Prism software. Data are presented as means ± SEM. Multiple group comparisons were analyzed using one-way and two-way ANOVA with the post hoc Bonferroni test. Unpaired t-test was used for comparison between two groups. p-values ≤0.05 were determined to be significant unless otherwise noted in legends.

## Acknowledgements

We thank the Systems Biology laboratory at the University of Texas, MD Anderson Cancer Center, for the initial analysis and for performing the microarray studies. We would also like to thank the metabolite analysis core at Baylor College of Medicine for performing the bile acid composition analysis. We thank Dr. Bhoomika Mathur for her initial help with the DDC experiment. The authors also thank Drs. Auinash Kalsotra and Stephanie Ceman for their comments and critiques during the preparation of this manuscript. We also thank Ms. Angela Major at Texas Children's' Hospital for histological preparation and analysis. The authors thank Dr. Lucas Li at the Roy Carver Metabolomics Core at the University of Illinois at Urbana-Champaign for Urea Cycle metabolite analysis. This work was supported in part by the NIDDK grant, R01 DK113080 (SA), Research Scholar Grant ACS 132640-RSG (SA), and UIUC start-up funds (SA)

## Additional information

### Competing interests

Pieter Dorrestein: Dr. Dorrestein is an advisor and holds equity in Cybele and Sirenas and a Scientific co-founder, advisor and holds equity to Ometa, Enveda, and Arome with prior approval by UC-San Diego and also consulted for DSM animal health in 2023. The other authors declare that no competing interests exist.

### Funding

| Funder | Grant reference number | Author |
|---|---|---|
| National Institute of Diabetes and Digestive and Kidney Diseases | DK113080 | Sayeepriyadarshini Anakk |
| American Cancer Society | 132640-RSG | Sayeepriyadarshini Anakk |
| University of Illinois Urbana-Champaign | start-up funds | Sayeepriyadarshini Anakk |

The funders had no role in study design, data collection and interpretation, or the decision to submit the work for publication.

### Author contributions

Megan E Patton, Sherwin Kelekar, Conceptualization, Data curation, Formal analysis, Validation, Investigation, Visualization, Methodology, Writing – original draft; Lauren J Taylor, Formal analysis, Validation, Investigation, Visualization, Methodology, Writing – review and editing, L.J.T. helped with urea cycle expression analysis; Angela E Dean, Formal analysis, Investigation, Methodology, A.E.D assisted with preparing samples for ERa-ChIP; Qianying Zuo, Data curation, Formal analysis, Validation, Investigation, Visualization, Methodology, Q.Z performed ERa-ChIP PCR analysis; Rhishikesh N Thakare, Data curation, Formal analysis, Validation, Investigation, Visualization, Methodology, R.N.T. analyzed bile acid composition from serum and urine; Sung Hwan Lee, Resources, Data curation, Formal analysis, Validation, Visualization, Methodology, Writing – review and editing, SHL performed all the HCC patient cohort analysis; Emily C Gentry, Data curation, Formal analysis, Validation, Investigation,

Visualization, Methodology, Writing – original draft, E.G.performed and analyzed untargeted metabolomics; Morgan Panitchpakdi, Data curation, Formal analysis, Validation, Investigation, Visualization, Methodology, M.P assisted with analysis of untargeted metabolomics; Pieter Dorrestein, Resources, Data curation, Formal analysis, Supervision, Validation, Investigation, Visualization, Methodology, Writing – original draft, P.D assisted with analysis and interpretation of untargeted metabolomics; Yazen Alnouti, Resources, Data curation, Formal analysis, Supervision, Validation, Investigation, Visualization, Methodology, Writing – original draft, Project administration, Y.A. analyzed and interpreted bile acid composition from serum and urine; Zeynep Madak-Erdogan, Resources, Data curation, Formal analysis, Supervision, Validation, Investigation, Visualization, Methodology, Project administration, Writing – review and editing, Z.M.E assisted with the analysis of ERa ChIP- and ERa-ChIP PCR data; Ju-Seog Lee, Resources, Data curation, Formal analysis, Validation, Investigation, Visualization, Methodology, Project administration, Writing – review and editing, J.S.L assisted with all the HCC patient cohort analysis and interpretations; Milton J Finegold, Resources, Data curation, Formal analysis, Supervision, Validation, Investigation, Visualization, Methodology, Writing – original draft, Writing – review and editing, M.J.F. evaluated and interpreted the liver histology; Sayeepriyadarshini Anakk, Conceptualization, Resources, Data curation, Supervision, Funding acquisition, Validation, Investigation, Visualization, Methodology, Writing – original draft, Project administration, Writing – review and editing

#### Author ORCIDs
Megan E Patton ⓘ https://orcid.org/0000-0002-2896-1415
Lauren J Taylor ⓘ https://orcid.org/0000-0001-9168-0870
Angela E Dean ⓘ https://orcid.org/0000-0002-9141-2590
Qianying Zuo ⓘ https://orcid.org/0000-0002-3288-2672
Sung Hwan Lee ⓘ https://orcid.org/0000-0003-3365-0096
Emily C Gentry ⓘ https://orcid.org/0000-0002-0016-8132
Yazen Alnouti ⓘ https://orcid.org/0000-0002-3995-3242
Milton J Finegold ⓘ https://orcid.org/0000-0002-2153-500X
Sayeepriyadarshini Anakk ⓘ https://orcid.org/0000-0003-2819-695X

#### Ethics

All studies were carried out in strict accordance as outlined in the Guide for the Care and Use of Laboratory Animals prepared by the National Academy of Sciences and published by the National Institutes of Health (National Institutes of Health publication 86-23, revised 1985). All of the animals were handled according to approved institutional animal care and use committee (IACUC) protocols (#22183) of the University of Illinois, Urbana Champaign.

Reviewer #1 (Public review): https://doi.org/10.7554/eLife.96783.4.sa1
Author response https://doi.org/10.7554/eLife.96783.4.sa2

## Additional files

#### Supplementary files
MDAR checklist
Supplementary file 1. The title is written in the word doc- Primer Sequences used.

#### Data availability
The gene expression data generated and used in this publication have been deposited in NCBI's Gene Expression Omnibus and are accessible through GEO Series accession number GSE151524 (https://www.ncbi.nlm.nih.gov/geo/query/acc.cgi?acc=GSE151524). All other data are included in the manuscript.

The following dataset was generated:

| Author(s) | Year | Dataset title | Dataset URL | Database and Identifier |
|---|---|---|---|---|
| Anakk S, Kim K, Lee JS, Jeong YS, Lee SH | 2020 | Analysis of tumor-bearing Farnesoid X Receptor and Small heterodimer Partner double knockout and control WT mice in both sexes | https://www.ncbi.nlm.nih.gov/geo/query/acc.cgi?acc=GSE151524 | NCBI Gene Expression Omnibus, GSE151524 |

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
