## [Editor Report · eLife Assessment]

This study provides **valuable** insights into the influence of sex on bile acid metabolism and the risk of hepatocellular carcinoma (HCC). The data to support that there are inter-relationships between sex, bile acids, and HCC in mice are **convincing**, although this is a largely descriptive study. Future studies are needed to understand the interaction of sex hormones, bile acids, and chronic liver diseases and cancer at a mechanistic level. Also, there is not enough evidence to determine the clinical significance of the findings given the differences in bile acid composition between mice and men.

---

## [Referee Report · Reviewer #1 (Public review)]

Liver cancer shows a high incidence in males than females with incompletely understood causes. This study utilized a mouse model that lacks the bile acid feedback mechanisms (FXR/SHP DKO mice) to study how dysregulation of bile acid homeostasis and a high circulating bile acid may underlie the gender-dependent prevalence and prognosis of HCC. By transcriptomics analysis comparing male and female mice, unique sets of gene signatures were identified and correlated with HCC outcomes in human patients. The study showed that ovariectomy procedure increased HCC incidence in female FXR/SHP DKO mice that were otherwise resistant to age-dependent HCC development, and that removing bile acids by blocking intestine bile acid absorption reduced HCC progression in FXR/SHP DKO mice. Based on these findings, the authors suggest that gender-dependent bile acid metabolism may play a role in the male-dominant HCC incidence, and that reducing bile acid level and signaling may be beneficial in HCC treatment. This study include many strengths: 1. Chronic liver diseases often proceed the development of liver and bile duct cancer. Advanced chronic liver diseases are often associated with dysregulation of bile acid homeostasis and cholestasis. This study takes advantage of a unique FXR/SHP DKO model that develop high organ bile acid exposure and spontaneous age-dependent HCC development in males but not females to identify unique HCC-associated gene signatures. The study showed that the unique gene signature in female DKO mice that had lower HCC incidence also correlated with lower grade HCC and better survival in human HCC patients. 2. The study also suggests that differentially regulated bile acid signaling or gender-dependent response to altered bile acids may contribute to gender-dependent susceptibility to HCC development and/or progression. 3. The sex-dependent differences in bile acid-mediated pathology clearly exist but are still not fully understood at the mechanistic level. Female mice have been shown to be more sensitive to bile acid toxicity in a few cholestasis models, while this study showed a male dominance of bile acid promotion of HCC. This study used ovariectomy to demonstrate that female hormones are possible underlying factors. Future studies are needed to understand the interaction of sex hormones, bile acids, and chronic liver diseases and cancer.

---

## [Author Response]

The following is the authors’ response to the previous reviews

**Reviewer #1 (Public review):**
Summary:Liver cancer shows a high incidence in males than females with incompletely understood causes. This study utilized a mouse model that lacks the bile acid feedback mechanisms (FXR/SHP DKO mice) to study how dysregulation of bile acid homeostasis and a high circulating bile acid may underlie the gender-dependent prevalence and prognosis of HCC. By transcriptomics analysis comparing male and female mice, unique sets of gene signatures were identified and correlated with HCC outcomes in human patients. The study showed that ovariectomy procedure increased HCC incidence in female FXR/SHP DKO mice that were otherwise resistant to agedependent HCC development, and that removing bile acids by blocking intestine bile acid absorption reduced HCC progression in FXR/SHP DKO mice. Based on these findings, the authors suggest that gender-dependent bile acid metabolism may play a role in the male-dominant HCC incidence, and that reducing bile acid level and signaling may be beneficial in HCC treatment.strengths:(1) Chronic liver diseases often proceed the development of liver and bile duct cancer. Advanced chronic liver diseases are often associated with dysregulation of bile acid homeostasis and cholestasis. This study takes advantage of a unique FXR/SHP DKO model that develop high organ bile acid exposure and spontaneous age-dependent HCC development in males but not females to identify unique HCC-associated gene signatures. The study showed that the unique gene signature in female DKO mice that had lower HCC incidence also correlated with lower grade HCC and better survival in human HCC patients. 2. The study also suggests that differentially regulated bile acid signaling or gender-dependent response to altered bile acids may contribute to gender-dependent susceptibility to HCC development and/or progression. 3. The sex-dependent differences in bile acidmediated pathology clearly exist but are still not fully understood at the mechanistic level. Female mice have been shown to be more sensitive to bile acid toxicity in a few cholestasis models, while this study showed a male dominance of bile acid promotion of HCC. This study used ovariectomy to demonstrate that female hormones are possible underlying factors. Future studies are needed to understand the interaction of sex hormones, bile acids, and chronic liver diseases and cancer.

We thank Reviewer 1 for their positive and thorough assessment of our manuscript

Weaknesses:(1) HCC shows heterogeneity, and it is unclear what tissues (tumor or normal) were used from the DKO mice and human HCC gene expression dataset to obtain the gene signature, and how the authors reconcile these gene signatures with HCC prognosis.

Mice studies: Aged DKO mice develop aggressive tumors (major and minor nodules, See Figure 1), and the entire liver is burdened with multiple tumor nodules. It is technically challenging to demarcate the tumor boundaries as most of the surrounding tissues do not display normal tissue architecture. Therefore, livers from age- and sexmatched wild-type C57/BL6 mice were used as control tissue. All the mice were inbred in our facility. Spatial transcriptomics and longitudinal studies are ongoing to collect tumors at earlier time points wherein we can differentiate tumor and non-tumor tissue.

Human Studies: We mined five separate clinical data sets. The human HCC gene expression comprised of samples from the (i) National Cancer Institute (NCI) cohort (GEO accession numbers, GSE1898 and GSE4024) and (ii) Korea, (iii) Samsung, (iv) Modena, and (v) Fudan cohorts as previously described (GEO accession numbers, GSE14520, GSE16757, GSE43619, GSE36376, and GSE54236). We have added a new supplemental table 4, giving details of these datasets. Depending on the cohort, they are primarily HCC samples- surgical resections of HCC, control samples, with some tumors and paired non-tumor tissues.

(2) The authors identified a unique set of gene expression signatures that are linked to HCC patient outcomes, but analysis of these gene sets to understand the causes of cancer promotion is still lacking. The studies of urea cycle metabolism and estrogen signaling were preliminary and inconclusive. These mechanistic aspects may be followed up in revision or future studies.

We agree. Experiments to elicit HCC causality and promotion are complex, given the heterogeneous nature of liver cancer. Moreover, the length of time (12 months) needed to spontaneously develop cancer in this DKO mouse model makes it challenging. As mentioned by the reviewer, mechanistic studies are ongoing, and longitudinal time course experiments are actively being pursued to delineate causality. Having said that, we mined the TCGA LIHC (The Cancer Genome Atlas Liver Hepatocellular Carcinoma) database to examine the expression of the individual urea cycle genes and found them suppressed in liver tumorigenesis (new Supplementary Figure 4). We also evaluated if estrogen receptor α (Erα) targets altered in DKO females (DKO_Estrogen) correlate with overall survival in HCC (new Supplementary Figure 6). We note that Er expression per se is reduced in males and females upon liver tumorigenesis. Also, DKO_Estrogen signature positively corroborated with better overall survival (new Supplementary Figure 6). These findings further bolster the relevance of urea cycle metabolism and estrogen signaling during HCC.

(3) While high levels of bile acids are convincingly shown to promote HCC progression, their role in HCC initiation is not established. The DKO model may be limited to conditions of extremely high levels of organ bile acid exposure. The DKO mice do not model the human population of HCC patients with various etiology and shared liver pathology (i.e. cirrhosis). Therefore, high circulating bile acids may not fully explain the male prevalence of HCC incidence.

We agree with this comment that our studies do not show bile acids can initiate HCC and may act as one of the many factors that contribute to the high male prevalence of HCC. This is exactly the reason why throughout the manuscript we do not write about HCC initiation. To clarify further, in the revised discussion of the manuscript, we have added a sentence to highlight this aspect, “while this study demonstrates bile acids promote HCC progression it does not investigate or provide evidence if excess bile acids are sufficient for HCC initiation.”

(4) The authors showed lower circulating bile acids and increased fecal bile acid excretion in female mice and hypothesized that this may be a mechanism underlying the lower bile acid exposure that contributed to lower HCC incidence in female DKO mice. Additional analysis of organ bile acids within the enterohepatic circulation may be performed because a more accurate interpretation of the circulating bile acids and fecal bile acids can be made in reference to organ bile acids and total bile acid pool changes in these mice.

As shown in this manuscript- we provide BA compositional analyses from the liver, serum, urine, and feces (Figures 5 and 6, new Supplementary Figure 8, Supplementary Tables 4 and 5). Unfortunately, we did not collect the intestinal tissue or gallbladders for BA analysis in this study. Separate cohorts of mice are being aged for future BA analyses from different organs within the enterohepatic loop. We thank you for this suggestion. Nevertheless, we have previously measured and reported BA values to be elevated in the intestines and the gall bladder of young DKO mice (PMC3007143).

**Reviewer #2 (Public review):**
Weaknesses:(1) The translational value to human HCC is not so strong yet. Authors show that there is a correlation between the female-selective gene signature and low-grade tumors and better survival in HCC patients overall. However, these data do not show whether this signature is more highly correlated with female tumor burden and survival. In other words, whether the mechanisms of female protection may be similar between humans and mice. In that respect, it would also be good to elaborate on whether women have higher fecal BA excretion and lower serum BA concentration.

The reviewer poses an interesting question to test if the DKO female-specific signatures are altered differently in male vs. female HCC samples. As we found the urea cycle and estrogen signaling to be protective and enriched in our mouse model, we tested their expression pattern using the TCGA-LIHC RNA-seq data. We found urea cycle genes and Erα transcripts broadly reduced in tumor samples irrespective of the sex (new Supplementary Figure 4 and Supplementary Figure 6), indicating that these pathways are compromised upon tumorigenesis even in the female livers.

While prior studies have shown (i) a smaller BA pool w synthesis in men than women (PMID: 22003820), we did not find a study that systematically investigated BA excretion between the sexes in HCC context. The reviewer is spot on in suggesting BA analysis from HCC and unaffected human fecal samples from both sexes. Designing and performing such studies in the future will provide concrete proof of whether BA excretion protects female livers from developing liver cancer. We thank you for these suggestions.

(2) The authors should perform a thorough spelling and grammar check.

We apologize for the typos, which have been fixed, and as suggested by the reviewer, we have performed a grammar check.

(3) There are quite some errors and inaccuracies in the result section, figures, and legends. The authors should correct this.

We apologize for the inadvertent errors in the manuscript, and we have clarified these inaccuracies in the revised version. Thank you.

**Reviewer#1 (Recommendations for the authors)**.(1) Figures 1A-F, This statement of altered liver steatosis needs to be further supported by measurement of liver triglycerides. Lower magnification images of Sirius red stain should be shown for better evaluation of liver fibrosis.

Unfortunately, we did not measure liver triglycerides and sirius red stained samples have faded, and lower magnification is unavailable at this juncture. We have modified our results accordingly.

We did not take the gross picture of WT female and DKO female livers in the same frame as shown below. Since the manuscript is focused on male and female differences in liver cancer incidence, we provided DKO male and female liver images as Figure 1D in the paper.

**Author response image 1. sa2fig1:** Gross liver images of a year-old WT and DKO mice which show prominent hepatocarcinogenesis in DKO male mice.

(2) Can the authors clarify if the gene transcriptomics was performed with normal or tumor tissues of DKO mice?

Gene transcriptomics were performed with the tumor tissue of DKO mice. We have previously published data from younger non tumor bearing DKO male mice (PMCID: PMC3007143).

(3) Supplementary Figure 3C. Could the authors confirm if this is F vs M or just DKO female since it does not seem to match the result description in the main text? It is better practice to indicate the sub-panels of the Supplementary Figures in the main text while describing the results.

As the reviewer correctly points out Supplementary Figure 3C is DKO F vs M signature not DKO_female signature and this has been clarified in the text. We have also included DKO_F data now to reduce the confusion.

(4) Figure 3. Legend, the data presented are not well explained in the Legend, especially the labeling and what is being presented and compared.

As suggested by the reviewer, we have modified the legend accordingly.

(5) Supplementary Table 4 does not contain total serum bile acid as described in the main text.

We agree with the reviewer. We provided primary and secondary BA concentrations, Supplementary Table 4 (currently Supplementary Table 5 in the revised version): Rows 20 and 21. but not their added total. We have modified the text accordingly.

(6) Method section: many experiments lack descriptions of details.

We have added details to the animal experimental design, ER ChIP-PCR, schematics of experiments are included within the main and supplemental figures, metabolomics and BA analysis have been expanded.

**Reviewer #2 (Recommendations For The Authors):**
General:(1) The authors are advised to do a thorough grammar and spelling check.

We have performed spelling and grammar check as suggested using an online platform Grammarly. Thank You.

Results:(1) Figure 1 o The authors should show in Figure 1D female WT and female DKO liver.

See Figure 1 added in our responses to point 1 of reviewer 1’s comment.

In the Figure legend, (A-E) should be replaced by (A+D).

Thank you. We have modified it accordingly.

The authors do not refer to 1J in the text, please add this reference.

Thank you for pointing it. We have referenced 1J in the text.

The description of 1H does not elaborate on the sex differences in ALT/AST levels, as this is the focus of the manuscript.

We have added a sentence to show that the injury markers are higher in DKO males, which is consistent with an advanced disease. Thanks.

The authors should use the correct nomenclature in Figure 1I/1J (gene vs protein and capitals vs non-capitals).

The Figure 1I and 1J show gene expression of Fxr and Shp and hence we used the non-capital italicized nomenclature. Thanks.

(2) Figure 2:The x-axis length is different in Figures 2A and 2B. Please correct to visualize the differences between males and females better.

The x axis length has been fixed as suggested. Thanks

(3) Figure 3:The authors should elaborate on how the patients were assigned to each gene signature. This is not fully clear.

The gene set obtained from the WT and DKO mice were used. The process used is shown as a schematic in Supplemental Fig 2C and the gene list is included in an excel sheet as Supplemental table 1.

We are curious how these data (F3A-C) would look when separating male and female human patients.

We performed an overall survival analysis with a subgroup of patients and provide it. We segregated the HCC cohort data on sex and age (>55 yr, since we assumed 55 as an age for menopause) and evaluated the DKO gene signature. Similar to the original figure 3, we find that irrespective of sex, and age, DKO FvsM gene signature corresponds with better overall survival in men and in women. These findings align with the combined analysis in overall survival shown in original Figure 3 of the manuscript, and therefore we did not modify it. If deemed necessary, we are happy to include the figure below to reviewers in the main manuscript.

**Author response image 2. sa2fig2:** Correlation of gene signatures obtained from WT and DKO mouse model with the survival data of HCC patients segregated by age and sex. The Kaplan Meier Survival graphs were generated based on WT and DKO transcriptome changes using five HCC clinical cohorts. Analysis of OS (Overall Survival) in patients ((A) Men and (B) Women) using the gene signatures representative of either male WT or male DKO, female WT or female DKO, and unique changes observed in female DKO mice but not in male DKO mice.

What was used as the control signature in Figure 3C? Please specify this.

For Figure 3C we compared the DKO_M signature to that of DKOF vs M signature. These genes are listed as an Excel Sheet (Supplementary Table 1).

The authors claim that DKO female mice display chronic cholestasis, similar to their male counterparts. Please refer to previous work or show the data.

Serum BA levels are elevated in DKO females are reported in supplementary table 5 and we find comparable hepatic BA composition in Figure 5 F.

(4) Figure 4: Labels for the x-axis are missing in Figure 4C. Please add legends or labels to the bars.

The x axis label is included in the top Serum BAs in (μM)

In Figure 4I, the percentage of input is quite low. An IgG control would show whether recruitment of ERalpha to the shown loci is significant above background levels. Also, ChIP on the OVX liver could serve as a negative control.

We did use IgG as control pull down and the signals above this background were considered. We have not performed this in OVX, which would be an excellent negative control for future studies. Thank You.

The results and legends refer to ChIP-qPCR, while methods only mention ChIP-seq.Please adapt.

We sincerely apologize for the mistake. We used published ChIP-seq to identify putative binding site and then performed ChIP PCR to validate it. We have clarified and rectified this error. Thank You.

Significance indications in the figure legend do not correspond with significance indications in the figure. Please explain the used significance symbols in the figure in the legend.

Thank You. The legends and their significance have been matched.

(5) Figure 5:Authors claim lowered total serum BA in females compared to males, and reference to Supplementary Table 4. However, these data are not provided, only percentages and ratios are displayed.

In the revised version, this has become Table 5. See response to the same concern noted by Reviewer 1, Point 5 above.

Figure 5D: Are sulphated BA also elevated in WT females? Please provide these data.

There is no significant urinary excretion of BAs in WT control animals. We have previously measured and found none. But under cholestatic conditions BAs are observed in urine. Therefore, sulphated BA levels were found only in the DKO mice.

Figure 5H: Is the fecal BA excretion in WT females also proportionally higher than in males? Please provide these data.

We were unable to perform the untargeted metabolomics profiling of WT fecal samples. When we measured for BAs in the feces, as expected very low conc were present irrespective of the sex (~0.01 μM) and we did not find any sex difference. Also, prior studies in 129SVJ strain exhibited comparable fecal excretion (PMC150802). We did not find any clinical studies that measured fecal BA between the sexes.

(6) Figure 6:References in the text of the result section to Figure 6 are wrong. The authors should change this.

Thank You. This has been rectified.

Significance indications in the legend do not correspond with significance indications in the figure. Please explain the used significance symbols in the figure in the legend.

Thank You. The legends and their significance have been matched.

(7) Supplemental Figure 3:Please adapt the title of this figure; the sentence is incorrect. The description of this figure is very poor.

We have modified the legend and the title of the Supplemental Figure 3 to make it more appropriate. Thanks

Please explain what the blue and red dots represent.

Each dot in blue and yellow indicate the Bayesian probability generated from our BCCP model.

What are the bold horizontal lines representing? Why are there no dots in some box plots? Please elaborate.

The box represents the interquartile range (IQR), encompassing the middle 50% of the data. The bottom and top edges correspond to the 25th and 75th percentiles, respectively, while the bold horizontal line indicates the median value.

The absence of visible dots in certain categories—particularly in higher CLIP and TNM stages—is due to the small number of patients, all of whom had similar Bayesian prediction probabilities. As these values cluster tightly around the median, the individual dots may be overlapped and hidden behind the median line.

The figure is not visually easy to understand, please reconsider the representation.

We hope the modified figure legends with the explanation of the lines and the points in the graphs increases the clarity and makes them acceptable.

Please add the DKO_female signature plot.

We have added these graph to Supplemental figure 3

(8) Supplemental 4A:Fold change at Z-score is missing. This should be added.

Thank you we have added this information

(9) Supplemental 5:The scale bar is missing. This should be included.

The figure is now supplemental figure 8 and the scale bar has been added.

Methods:(1) Did the authors use ChIP-sequencing or ChIP-qPCR? Please describe the correct method.

We apologize for the error. We have used ChIP-PCR and rectified it in our methods and in our response to a figure 4 query.

(2) It is unclear how the mouse model was generated. Please refer to earlier publications.

The mice were generated in house at UIUC, and we have added this sentence to the Methods section. The original reference has been cited in the text (PMCID: PMC3007143).

Discussion:(1) The authors claim in the discussion: 'consistently higher recruitment of ER to the classical BA synthetic genes ...' This is not shown in Figure 4I, only ER recruitment to Cyp7a1 is significantly higher in females. Please rephrase.

We agree and we have modified the sentence Cyp7A1 accounts for ~75% of BA synthesis and is a rate-limiting gene in the classical BA synthesis pathway.

(2) The authors could make their statements stronger if they could elaborate on whether women have more fecal BA excretion, and if there are differences in serum BA concentration in HCC between male and female patients.

Unfortunately, we were unable to find clinical studies with appropriate controls which examined and reported serum BA in HCC in a sex specific manner.

In addition, to understand whether the female-specific protections in humans are similar to mice, it would be nice to show correlations of the female-specific mouse signature with male and female liver signatures.

At this time, we do not have large n numbers of control or precancerous early-stage patient datasets from both sexes to make such comparisons. Nevertheless, there is translational relevance of these sex-specific signature. Figure 2 included in the reviewer response shows that DKO male signature correlates with poor overall survival in males, whereas neither DKO male nor DKO female signature predict outcome in females. In contrast, DKO female-specific gene signature (DKOFvsM) correlates with better overall survival in both men and in women.

(3) The authors state in the discussion: 'Currently we do not know how to reconcile this data other than indicating a potential ER independent mechanism.' We do not understand the reasoning behind this statement. Please clarify.

We find that increased Erα expression in DKO coincides with CA-mediated suppression of BA synthesis genes in the absence of Fxr and Shp. But we also noticed that in OVX DKO mice, Erα expression is blunted, and so is basal BA synthesis gene expression. Putting together these data, it is intriguing that Erα expression correlates both positively and negatively with BA synthesis genes. To reconcile these contrasting results, we have written the following sentence in the discussion.

“These findings suggest Erα expression is linked to both positive and negative regulation of BA synthesis genes. But we do not know how ER elicits these differential effects on BA synthesis.”